# Tensile and Flexural Properties of Silica Nanoparticles Modified Unidirectional Kenaf and Hybrid Glass/Kenaf Epoxy Composites

**DOI:** 10.3390/polym12112733

**Published:** 2020-11-18

**Authors:** Napisah Sapiai, Aidah Jumahat, Mohammad Jawaid, Mohamad Midani, Anish Khan

**Affiliations:** 1Faculty of Mechanical Engineering, Universiti Teknologi MARA, Shah Alam, Selangor 40450, Malaysia; napisah@uitm.edu.my; 2Institute for Infrastructure Engineering Sustainable and Management, Universiti Teknologi MARA, Shah Alam, Selangor 40450, Malaysia; 3Department of Biocomposite Technology, Institute of Tropical Forestry and Forest Products, Universiti Putra Malaysia, UPM Serdang, Selangor 43400, Malaysia; 4Wilson College of Textiles, NC State University, Raleigh, NC 27606, USA; msmidani@ncsu.edu; 5Center of Excellence for Advanced Materials Research, King Abdulaziz University, P.O. Box. 80203, Jeddah 21589, Saudi Arabia; anishkhan97@gmail.com

**Keywords:** nanosilica, kenaf fiber, glass fiber, polymer composites, tensile properties, flexural properties

## Abstract

This paper investigates the influence of silica nanoparticles on the mechanical properties of a unidirectional (UD) kenaf fiber reinforced polymer (KFRP) and hybrid woven glass/UD kenaf fiber reinforced polymer (GKFRP) composites. In this study, three different nanosilica loadings, i.e., 5, 13 and 25 wt %, and untreated kenaf fiber yarns were used. The untreated long kenaf fiber yarn was wound onto metal frames to produce UD kenaf dry mat layers. The silane-surface-treated nanosilica was initially dispersed into epoxy resin using a high-vacuum mechanical stirrer before being incorporated into the UD untreated kenaf and hybrid woven glass/UD kenaf fiber layers. Eight different composite systems were made, namely KFRP, 5 wt % nanosilica in UD kenaf fiber reinforced polymer composites (5NS-KFRP), 13% nanosilica in UD kenaf fiber reinforced polymer composites (13NS-KFRP), 25 wt % nanosilica in UD kenaf fiber reinforced polymer composites (25NS-KFRP), GKFRP, 5 wt % nanosilica in hybrid woven glass/UD kenaf fiber reinforced polymer composites (5NS-GKFRP), 13 wt % nanosilica in hybrid woven glass/UD kenaf fiber reinforced polymer composites (13NS-GKFRP) and 25 wt % nanosilica in hybrid woven glass/UD kenaf fiber reinforced polymer composites (25NS-GKFRP). All composite systems were tested in tension and bending in accordance with ASTM standards D3039 and D7264, respectively. Based on the results, it was found that the incorporation of homogeneously dispersed nanosilica significantly improved the tensile and flexural properties of KFRP and hybrid GKFRP composites even at the highest loading of 25 wt % nanosilica. Based on the scanning electron microscopy (SEM) examination of the fractured surfaces, it is suggested that the silane-treated nanosilica exhibits good interactions with epoxy and the kenaf and glass fibers. Therefore, the presence of nanosilica in an epoxy polymer contributes to a stiffer matrix that, effectively, enhances the capability of transferring a load to the fibers. Thus, this supports greater loads and improves the mechanical properties of the kenaf and hybrid composites.

## 1. Introduction

Kenaf fiber is one of the natural fibers that always receives much attention due to its potential as a natural-based reinforcement material in the composites industry. Kenaf fibers are extracted from the stem of the kenaf (*Hibiscus cannabinus*) plant belonging to the Malvaceace family [1,2,3,4]. Kenaf is an annual and short-day plant; it can grow to over 3–5 m tall with a woody base within 3–4 months [1,2,3,4]. Due to its high specific strength, low density, renewability, biodegradability and sustainability, kenaf fibers have also been touted as an alternative reinforcement material to replace glass fiber. Kenaf fibers have a tensile strength of 223–930 MPa and a modulus of 14.5–73 GPa, which are comparable to those of E-glass fibers that have a tensile strength of 1000 MPa and a modulus of 70–76 GPa [5]. Kenaf fiber composites have potential applications in many composites industries. They can be used as thermal insulation materials, sound-damping and vibration-absorbing materials, construction and building materials, food containers, coarse cloth and fabric, oil- and liquid-absorbent materials, etc. [4]. However, the strength, brittleness, toughness and stiffness properties of kenaf fibers when combined with matrix resin limit their applications. The processing and fabrication of kenaf fiber composites lead to the fracture and damage of the composite structures. As such, kenaf fiber composites are always modified with other materials called fillers [1,2,5,6,7,8,9,10]. The combination of kenaf fiber with fillers or its hybridization with other types of reinforcing fibers could be very cost-effective and enhance the mechanical performance of the kenaf fiber composites. The design of these hybrid composites could be tailored or engineered to suit the desired properties and applications.

Silica has received increased attention in materials sciences as a reinforcing nanofiller due to its high elastic modulus (70 GPa), high specific surface areas (50–380 m^2^/g), high thermal stability (1200 °C), low density (1.8 g/cm^3^), low thermal expansion coefficient and good abrasion resistance. Silica is composed of silicon dioxide (SiO_2_) and exists in crystalline and amorphous forms. Silica in the form of spherical amorphous nanoparticles with a range from 5 to 200 nm has become a vital area of research in FRP composites. A few researchers identified significant enhancement of the mechanical and thermal properties of FRP composites containing well-distributed nanosilica. Nayak et al. [11] and Jena et al. [12] proved that the addition of nanosilica/nanoclay and nanosilica/graphene in glass-fiber/epoxy composites offered a lot of benefits; in particular, it improved the flexural strength properties. The authors concluded that the addition of nanosilica strengthened the adhesion properties of glass/epoxy interfaces. In woven-carbon-fabric/epoxy composites studied by Moghimi et al. [13], the inclusion of nanosilica/carbon nanotubes improved the tensile strength, tensile modulus and tribological properties of the composites. This was due to good dispersion of the nanofiller that strengthened the surrounding matrix, interfacial bonding and resin adhesion. In addition, the inclusion of nanosilica in phenolic resin also contributed to significant improvement of mechanical properties. For example, in the study by Mirzapour et al. [14], the bending strength of the nanosilica/carbon-fiber/phenolic nanocomposite was increased by about 13% by adding 3 wt % nanosilica. Khodadadi et al. [15] found that the addition of nanosilica improved the ballistic performance of woven Kevlar fabric impregnated with a colloidal shear thickening fluid (STF). The force required to pull the Kevlar yarn out from the fabric impregnated with STF increased with increasing nanosilica loading. In addition, the impact resistance performance also improved as nanosilica loading in STF increased from 15 to 35 wt %; however, the effectiveness decreased with further loading of nanosilica.

The addition of nanosilica in FRP composites was also extensively studied by Jumahat and coworkers [2,16,17,18,19,20]. It was found that the addition of nanosilica significantly improved the mechanical properties of the carbon fiber reinforced polymer (CFRP). As reported in [18], excellent improvement of the compressive properties was obtained for CFRP composites modified with nanosilica. It was reported that the presence of 25 wt % nanosilica improved the compressive modulus and strength by more than 30% and 70%, respectively. The results showed that the presence of spherical silica nanoparticles stiffened the epoxy matrix and offered better lateral support to the carbon fiber, thus improving the compressive properties [18]. In another study, the presence of nanosilica increased tensile strength up to 60% and improved ductility and toughness of basalt fiber reinforced composites [20]. The incorporation of silica nanoparticles was also found to improve the open hole compressive strength and stiffness of glass fiber reinforced polymer (GFRP), Kevlar fiber reinforced polymer (KFRP) and hybrid GFRP/KFRP composites. This was due to the good adhesion bonding of the matrix and fibers and the reduced barreling and crimping of Kevlar fibers [16]. In addition, the exploration of the addition of nanosilica to different FRPs such as glass fiber [17], Kevlar fiber [16], kenaf fiber [2] and basalt fiber [20] also revealed significant enhancement of mechanical, thermal and tribological properties. However, it is interesting to note that there were contrasting findings when nanosilica was added to the silane, alkaline and acid surface-treated kenaf fiber composites. The addition of nanosilica resulted in a significant reduction in the longitudinal tensile properties of epoxy composites reinforced with treated kenaf fibers. Based on the study reported in [2], it can be concluded that the alkaline, acid and silane surface treatment processes removed or diminished the lignin structure of the kenaf fiber that protects the microfibril microstructure. These uncovered microfibrils caused very high resin absorption rate characteristics of the kenaf fiber and thus induced a fiber wetting problem. This wetting problem caused poor interfacial bonding between the fiber and the matrix, which led to a reduction in load transfer capability, premature failure and fiber pull-out of the kenaf composites [2]. The viscosity of the resin was higher with the presence of nanosilica, which worsened the mechanical properties of the surface-treated kenaf composites. As a result, based on the reported study in [2], the addition of nanosilica significantly reduced mechanical properties of the treated kenaf fiber composites. Therefore, this study focuses on the effect of nanosilica inclusion and the effect of glass fiber hybridization on the tensile and flexural properties of untreated kenaf fiber reinforced polymer composites.

## 2. Materials and Methods

### 2.1. Materials

The kenaf fiber yarn as shown in Figure 1a was supplied by Innovative Pultrusion Sdn. Bhd, Negeri Sembilan, Malaysia, and the woven glass fiber as shown in Figure 1b was provided by Vistec Technology, Selangor, Malaysia. The epoxy resin, MIRACAST 1517 was purchased from Miracon Sdn. Bhd, Selangor, Malaysia. The nanosilica, Nanopox F400, was supplied by Nanoresins AG, Geesthacht, Germany. Nanopox F400 consists of surface-modified synthetic SiO_2_ nanospheres of 40 wt % in the DGEBA-based epoxy resin. The nanosilica had an average particle size of 20 nm and was synthesized from an aqueous sodium silicate solution.

### 2.2. Fabrication of KFRP and Hybrid GKFRP Composites

The KFRP and hybrid GKFRP composite laminates were fabricated using a combination of filament winding and hand lay-up technique. The nanomodified resins were prepared by mixing the epoxy resin and hardener with a ratio of 100:30. Before that, the 5, 13 and 25 wt % nanosilica were first dispersed homogenously throughout the epoxy resin. This homogeneous dispersion was observed using transmission electron microscopy (TEM) as shown in Figure 2. The nanomodified resin was stirred using a high-vacuum mechanical stirrer for 1 h to reduce the entrapped air. Figure 3 shows the example of unidirectional KFRP composite laminates, and Table 1 describes the designation of KFRP and hybrid GKFRP composite systems with three different nanosilica contents.

### 2.3. Mechanical Testing

The tensile and flexural properties of kenaf and hybrid kenaf/glass composites were measured using a 100 kN Instron Universal Testing Machine (Norwood, MA, U.S.A). The tensile test was conducted at 2 mm/min crosshead speed on the tensile specimens of dimensions 250 × 15 mm. The tests were conducted in accordance with ASTM standard D3039. Flexural properties of kenaf and hybrid kenaf/glass composites were determined using ASTM D7264. The flexural specimens of dimensions 80 × 13 mm were prepared. A 16:1 span-to-depth ratio of samples was used, and the crosshead speed of the flexural tests was set to 2 mm/min. In order to ensure repeatability and accuracy of the results, five samples were tested for each type of composite system and for each type of mechanical test.

### 2.4. Microstructure Evaluation

The morphological structures of the fractured surfaces after tensile tests were observed using a Hitachi TM 3000 (Illinois, U.S.A) scanning electron microscope (SEM). After the tensile test, the fractured samples were cut into heights of 10–15 mm. These samples were then sputter-coated with gold for about 45 s using an SC7620 QUORUM (East Sussex, U.K.) sputter coater machine to produce electrically conductive specimens and to prevent charge build-up by electron absorption.

## 3. Results and Discussion

Based on the previous work reported in [2,19], it was found that the mechanical performance of the surface-treated kenaf fiber composites was low when compared to that of the untreated kenaf composites. In addition, the presence of nanosilica in the surface-treated kenaf composites resulted in reduction in the mechanical performance. Therefore, this research work focused on the properties of epoxy polymer composites reinforced with untreated long kenaf fiber yarn. The effects of nanosilica and hybridization with woven glass on the tensile and flexural properties of kenaf composites were studied. The influence of three different nanosilica loadings, i.e., 5, 13 and 25 wt %, on the mechanical properties of kenaf and hybrid kenaf/glass composites were analyzed and are discussed in the following subsections (Section 3.1 and Section 3.2).

### 3.1. Tensile Properties

Figure 4a illustrates the tensile properties of kenaf composites (KFRP) with inclusion of three different nanosilica contents, and Figure 4b shows the example of fractured samples. It was found that kenaf composites with nanosilica inclusion (nanosilica-modified KFRP) possessed better tensile properties when compared with the pure kenaf composites (KFRP). The tensile properties of the nanosilica-modified KFRP composites increased with increasing nanosilica content. The tensile modulus and tensile strength increased by 7.88% and 18.9%, respectively, when the kenaf composite was incorporated with 5 wt % nanosilica (5NS-KFRP). The 13NS-KFRP improved by 57.98% for tensile strength and 43.30% for tensile modulus. There were remarkable findings in the 25NS-KFRP composite as the tensile strength and modulus increased by 122.23% and 100.00%, respectively. The fractured surface morphology of the nanosilica-modified KFRP composites, as illustrated in Figure 5, clearly shows the nanosilica-modified epoxy resin is well wetted in the kenaf fiber composites, as there was no particle agglomeration. This, therefore, indicates that the silica nanoparticle was homogeneously distributed in the composite system.

In general, the SEM micrographs show that the failure mechanisms involved in the kenaf composites after they were loaded in tension were matrix cracking, fiber debonding and fiber pull-out. The pure KFRP composite system as shown in Figure 5a demonstrates that epoxy resin surrounded kenaf fiber yarn without proper diffusion into the spool or inside the fiber yarn. Moreover, the fiber debonding and large matrix cracking between the yarn kenaf fiber and the unmodified matrix indicate slightly weak adhesion bonding of the pure kenaf composite. However, with nanosilica inclusion (Figure 5b–d), the modified resin diffuses inside the yarn kenaf and creates good adhesion bonding. Surface-treated nanosilica acted as a bridging mechanism that enhanced the interaction between the kenaf and the epoxy resin, thus improving the mechanical properties (tensile and flexural properties) of the composites. Almost similar results and reasons were reported in the literature [6,11,12,13,14,15,16,17,18]. It was reported that nanosilica had good dispersion, improved adhesion bonding of the matrix and fibers, reduced fiber barrelling and crimping and offered a better lateral support to the fiber.

Based on this study, the addition of nanosilica in untreated kenaf fiber yarn significantly improved the tensile properties of the composites. This finding is in contrast to the results found in [2] for the surface-treated kenaf fiber. It was reported that the inclusion of nanosilica in NaOH-treated kenaf fiber composites reduced the longitudinal tensile properties [2]. This may be due to the presence of a functional group that is incompatible with the surface-modified nanosilica after the NaOH treatment on the kenaf fiber. There are a lot of reports from previous researchers on fiber-treatment-caused degradation or reduction of the fiber properties [21]. For example, Pickering et al. reported that the NaOH treatment of hemp fiber reduced K_IC_ by increasing its crystallinity. Osumani et al. [22] reported that the maximum value of the tensile modulus of the composite was found for the fiber treated with 5% NaOH concentration. The composites made of fiber surface-treated with an NaOH concentration greater than 5% showed a noticeable reduction in mechanical properties. This was possibly due to the fiber being damaged by excessive delignification [22].

Figure 6a demonstrates the tensile properties of the three different nanosilica contents in hybrid kenaf/glass composites (NS-GKFRP) when compared with the pure GKFRP system. Figure 6b shows the example of fractured hybrid composites samples after the tensile tests. In general, as shown in Figure 6a, it was found that the tensile properties of NS-GFRP increased with increasing nanosilica content. The tensile strength improved by 12.25%, 65.95% and 94% for 5NS-GFRP, 13NS-GFRP and 25NS-GFRP composites, respectively. Meanwhile, the tensile modulus improved by 23.71, 93.05 and 126.38 for 5NS-GFRP, 13NS-GFRP and 25NS-GFRP composites, respectively, as compared with the pure GKFRP composite system. Similar results were also reported by Norhashidah et al. [19] when nanosilica was added to the hybrid composite system of glass fiber/NaOH-treated kenaf composites. For this case, the hybridization of kenaf fibers with glass fiber improves the tensile properties of the kenaf composites.

### 3.2. Flexural Properties

Figure 7 and Figure 8 show the flexural test results of nanosilica-modified KFRP composites compared with KFRP composites and nanosilica-modified GKFRP compared with GKFRP composites, respectively. Figure 7b and Figure 8b show the example of fractured samples of KFRP and GKFRP composites, respectively, after being loaded in three-point bending. As seen in Figure 7a, the results suggest the benefits of the addition of nanosilica on flexural strength and flexural modulus for the KFRP composites. The flexural strength increased by 17.80%, 24.5% and 79.49% for 5NS-KFRP, 13NS-KFRP and 25NS-KFRP composites, respectively. Flexural modulus also showed improvements of 35.32%, 43.19% and 84.64% for 5NS-KFRP, 13NS-KFRP and 25NS-KFRP composites, respectively, when compared with KFRP.

For the hybrid composite systems, as shown in Figure 8a, the flexural strength and flexural modulus also increased with increasing nanosilica content. The recorded flexural strengths for the 5NS-GKFRP, 13NS-GKFRP and 25NS-GKFRP composites were about 119.05, 139.89 and 186.33 MPa and represented increases of 3.37%, 20.90% and 61.03% when compared with GFRP, respectively. The flexural modulus was improved by about 43.17% for 5NS-GKFRP, 71.16% for 13NS-GKFRP and 118.60% for 25NS-GKFRP. It was concluded that flexural properties for both KFRP and hybrid GKFRP improved with the inclusion of nanosilica. This behavior was expected due to matrix modification with nanosilica, where the fiber develops a strong interfacial bonding with the nanosilica-modified matrix resulting in higher flexural strength and modulus.

In a flexural test, there is a combination of tensile and compressive stress; therefore, the flexural properties of the composites depend on both fibers and matrix resin. For example, flexural modulus of the KFRP and hybrid GKFRP composites increased as the nanosilica content increased, which was due to nanosilica having a high elastic modulus as compared with pure epoxy resin. As such, combining epoxy and nanosilica boosted the elastic modulus for nanosilica-modified epoxy resin, which made it comparable to kenaf and glass fibers. The elastic moduli of epoxy resin, nanosilica, kenaf fibers and glass fibers are 3–6 GPa, 70 GPa, 14.5–73 GPa and 70–76 GPa, respectively, as reported by other researchers [2,5,23]. Furthermore, the presence of nanosilica not only improved adhesion bonding of the matrix and fibers, but also introduced additional mechanisms of energy absorption during compression, which gave a higher resistance against deformation [16,18]. Table 2 illustrates a summary of tensile and flexural properties for both the KFRP and hybrid GKFRP composites with inclusion of three different silica nanoparticle contents.

## 4. Conclusions

The effects of nanosilica on the tensile and flexural properties of the KFRP and hybrid GKFRP composites were successfully investigated. The results indicate that the inclusion of silica nanoparticles improved the tensile and flexural properties of KFRP and hybrid GKFRP composites. It is suggested that well-dispersed nanosilica particles within the epoxy matrix improved the adhesion bonding between the epoxy matrix, the silica nanofiller and the glass and kenaf fibers, which then led to improvements in tensile strength and tensile modulus. In addition, it is also concluded that the addition of treated silica to untreated kenaf and hybrid GKFRP composite systems significantly improved the flexural strength and flexural modulus. This may be due to the stiffer and tougher properties of the nanosilica-modified matrix introducing additional mechanisms of energy absorption during the bending test. Hence, this provided a higher resistance against deformation when the kenaf and hybrid composites were subjected to transverse load.

## Figures and Tables

**Figure 1 polymers-12-02733-f001:**
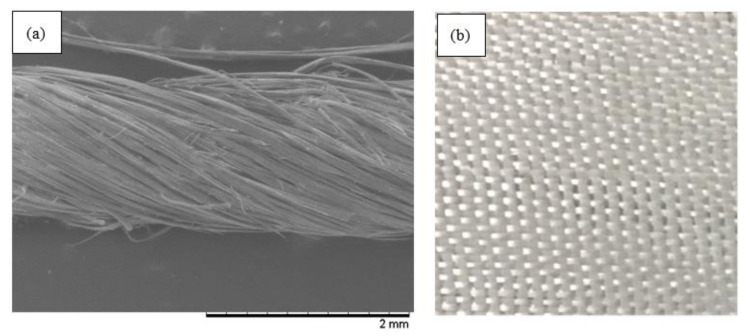
(**a**) Kenaf fiber in yarn form, viewed under scanning electron microscopy (SEM) at magnification of 50×. (**b**) Glass fiber woven roving (CRW200).

**Figure 2 polymers-12-02733-f002:**
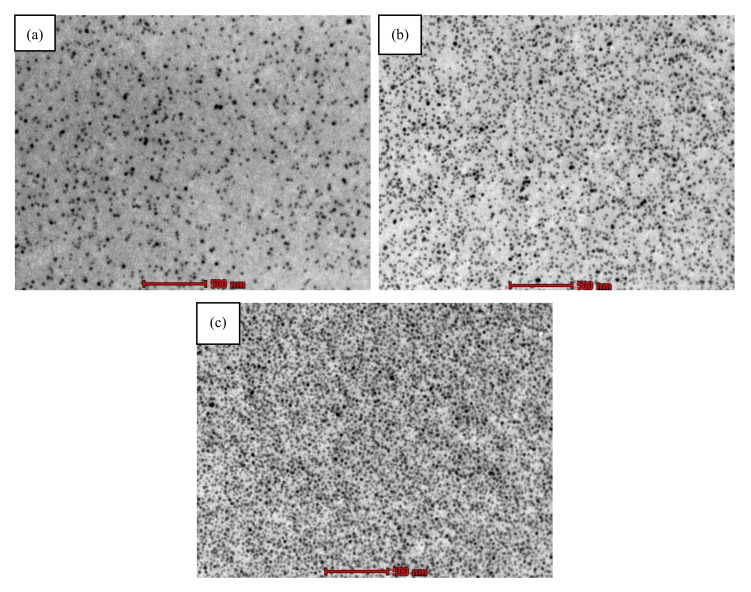
Nanosilica-modified epoxy, viewed under transmission electron microscopy (TEM) at magnifications of 20,500× for (**a**) 5 wt %, (**b**) 13 wt % and (**c**) 25 wt %.

**Figure 3 polymers-12-02733-f003:**
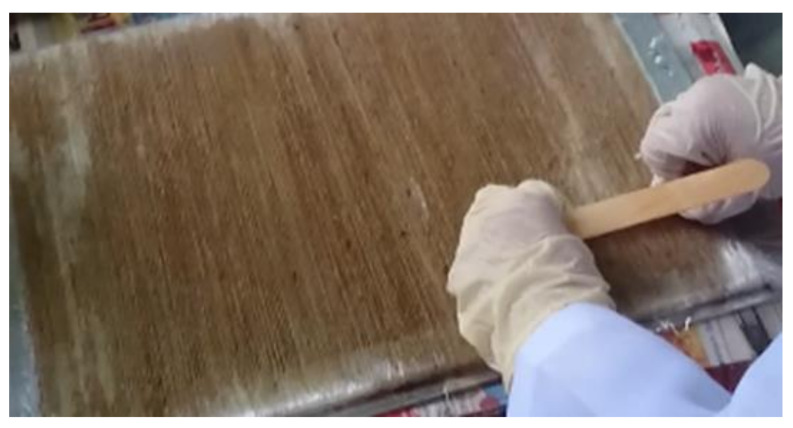
The example of a unidirectional kenaf fiber reinforced polymer (KFRP) composite sample after the curing process.

**Figure 4 polymers-12-02733-f004:**
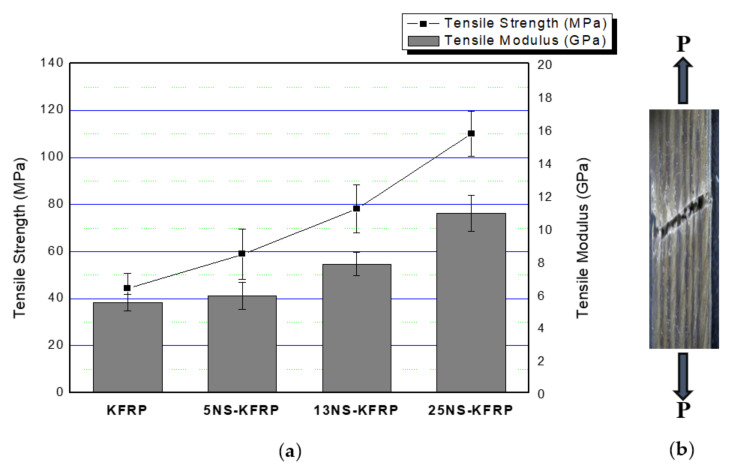
(**a**) Tensile properties of nanosilica-modified KFRP composites (5 wt % nanosilica in UD kenaf fiber reinforced polymer composites (5NS-KFRP), 13 wt % nanosilica in UD kenaf fiber reinforced polymer composites (13NS-KFRP) and 25 wt % nanosilica in hybrid woven glass/UD kenaf fiber reinforced polymer (25NS-KFRP)) compared with KFRP. (**b**) Example of fractured samples after tensile test.

**Figure 5 polymers-12-02733-f005:**
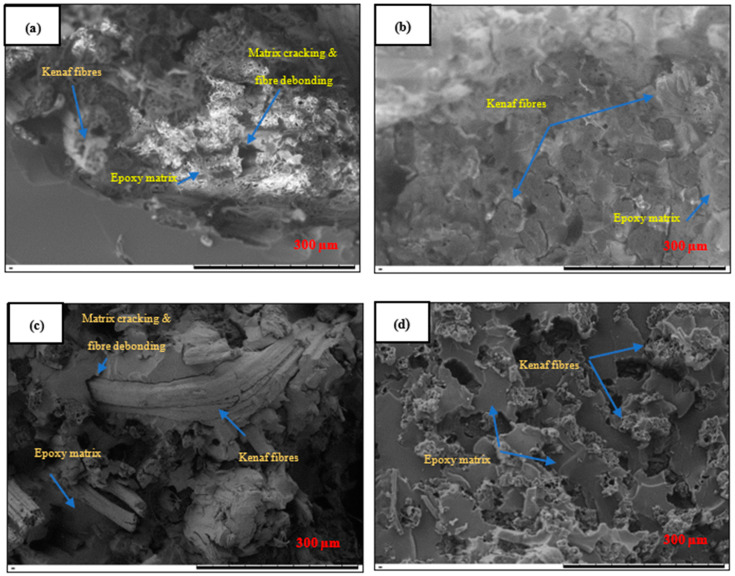
Morphological structures of the fractured surfaces of (**a**) KFRP, (**b**) 5NS-KFRP, (**c**) 13NS-KFRP and (**d**) 25NS-KFRP after they were subjected to the tensile test.

**Figure 6 polymers-12-02733-f006:**
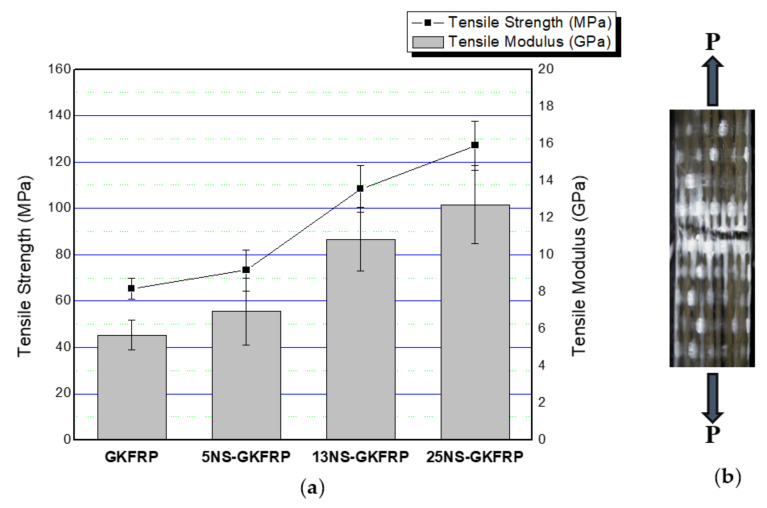
(**a**) Tensile properties of nanosilica-modified GKFRP (5 wt % nanosilica in hybrid woven glass/UD kenaf fiber reinforced polymer composites (5NS-GKFRP), 13 wt % nanosilica in hybrid woven glass/UD kenaf fiber reinforced polymer composites (13NS-GKFRP) and 25 wt % nanosilica in hybrid woven glass/UD kenaf fiber reinforced polymer composites (25NS-GKFRP)) compared with GKFRP. (**b**) Example of fractured GKFRP samples after tensile test.

**Figure 7 polymers-12-02733-f007:**
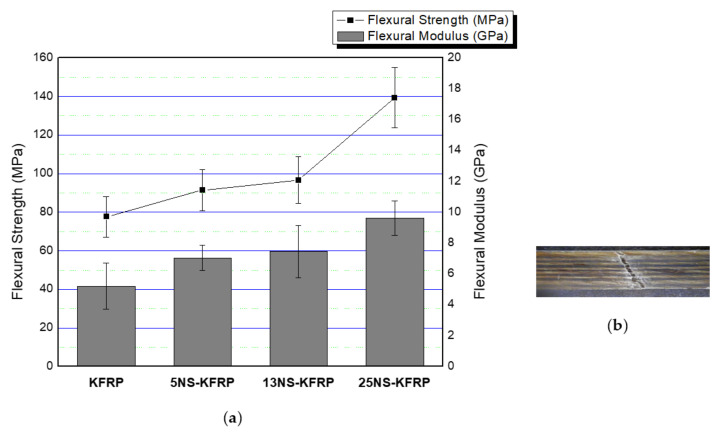
(**a**) Flexural properties of nanosilica-modified KFRP (5NS-KFRP, 13NS-KFRP and 25NS-KFRP) compared with KFRP. (**b**) Example of fractured KFRP samples after flexural test.

**Figure 8 polymers-12-02733-f008:**
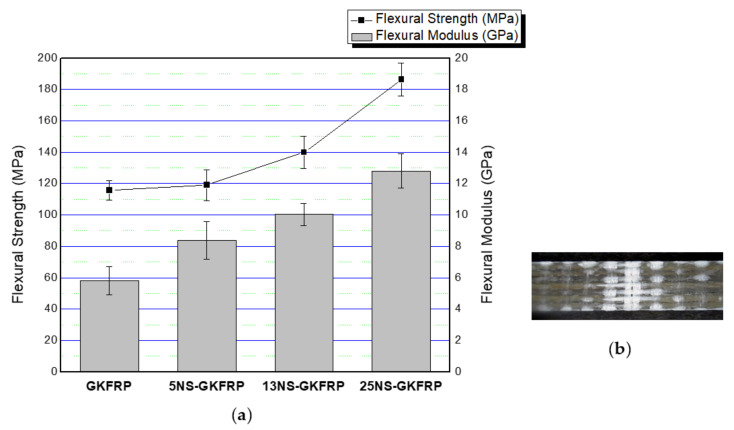
(**a**) Flexural properties of nanosilica-modified GKFRP (5NS-GKFRP, 13NS-GKFRP and 25NS-GKFRP) compared with GKFRP. (**b**) Example of fractured GKFRP samples after flexural test.

**Table 1 polymers-12-02733-t001:** The designation of unidirectional (UD) KFRP and hybrid woven glass/UD kenaf fiber reinforced polymer (GKFRP) composites with three different nanosilica loadings.

Composition of UD Kenaf and Hybrid Kenaf/Glass Composites	Designation
Unidirectional (UD) kenaf fiber reinforced polymer composites	KFRP
5 wt % nanosilica in UD kenaf fiber reinforced polymer composites	5NS-KFRP
13 wt % nanosilica in UD kenaf fiber reinforced polymer composites	13NS-KFRP
25 wt % nanosilica in UD kenaf fiber reinforced polymer composites	25NS-KFRP
Hybrid woven glass/UD kenaf fiber reinforced polymer composites	GKFRP
5 wt % nanosilica in hybrid woven glass/UD kenaf fiber reinforced polymer composites	5NS-GKFRP
13 wt % nanosilica in hybrid woven glass/UD kenaf fiber reinforced polymer composites	13NS-GKFRP
25 wt % nanosilica in hybrid woven glass/UD kenaf fiber reinforced polymer composites	25NS-GKFRP

**Table 2 polymers-12-02733-t002:** Tensile and flexural properties of KFRP and hybrid GKFRP with inclusion of nanoparticles.

Composites	Tensile Properties		Flexural Properties	
	Tensile Strength (MPa)	Tensile Modulus (GPa)	Flexural Strength (MPa)	Flexural Modulus (GPa)
KFRP	49.48 ± 6.45	5.45 ± 0.52	77.63 ± 10.45	5.21 ± 0.50
5NS-KFRP	58.84 ± 10.80	5.88 ± 0.81	91.45 ± 10.83	7.05 ± 0.76
13NS-KFRP	78.17 ± 10.14	7.81 ± 0.69	96.67 ± 12.12	7.46 ± 1.71
25NS-KFRP	109.96 ± 9.61	10.9 ± 1.12	139.34 ± 15.58	9.62 ± 1.10
GKFRP	65.29 ± 9.16	5.61 ± 0.37	115.71 ± 6.27	5.86 ± 0.12
5NS-GKFRP	73.29 ± 8.83	6.94 ± 1.81	119.05 ± 9.82	8.39 ± 1.22
13NS-GKFRP	108.35 ± 10.11	10.83 ± 1.72	139.89 ± 10.14	10.03 ± 0.73
25NS-GKFRP	127.14 ± 10.62	12.7 ± 2.14	186.33 ± 10.56	12.81 ± 1.11

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
