# Peer review of "Tensile and Flexural Properties of Silica Nanoparticles Modified Unidirectional Kenaf and Hybrid Glass/Kenaf Epoxy Composites"

_polymers, 2020, doi:10.3390/polym12112733_

Round 1
Reviewer 1 Report
The authors aim to the effect of nanosilica inclusion on tensile and flexural properties of the KFRP and hybrid GKFRP 252 composites. The work of this paper is practical and logical, however, there are some problems to be further improved as well: 1. This paper only provides the experimental results as shown in Fig. 4, Fig.5, Fig. 6 and Fig. 7 to support the conclusion. Simple test results do not constitute a scientific paper. So, the additional arguments are needed to explain the improvement effect of silica nanoparticle on mechanical. What are the reasons for this improvement?The fracture morphology, internal failure characteristics or the structure of the material can be provided to explain these question. 2. Modify the superscript or subscript of some units, such as units in Line 51 and Line 52.Author Response
|
No |
Comments/ Recommendations/ Suggestions/ Queries from the Reviewer |
Feedback from the Authors |
|
1. |
This paper only provides the experimental results as shown in Fig. 4, Fig.5, Fig. 6 and Fig. 7 to support the conclusion. Simple test results do not constitute a scientific paper. So, the additional arguments are needed to explain the improvement effect of silica nanoparticle on mechanical. What are the reasons for this improvement?The fracture morphology, internal failure characteristics or the structure of the material can be provided to explain these question.
|
Thank you for the comments and recommendations given by the reviewer.
The additional argument has been added. Figure 5 showing the SEM micrographs of the fractured surface of nanosilica reinforced kenaf composites samples. The image of fractured samples and the discussion on fracture mechanisms involved during the test have also been added. The text and explanation related to Figures 4, 5, 6, 7 and 8 and Table 2 have also been improved in Section 3 in order to further clarify the significance of this study, the analysis of the results together with the fracture mechanisms involved and to compare with those reported in the literature. (refer to pages 5-10, in new line 208-362)
|
|
2. |
Modify the superscript or subscript of some units, such as units in Line 51 and Line 52.
|
in Line 51 and Line 52, superscript or subscript have been modified. (refer to page 2, in new line 58-60)
|
Reviewer 2 Report
The paper was generally well written and can be accepted in the present form
Author Response
Reviewer#2
|
No |
Comments/ Recommendations/ Suggestions/ Queries from the Reviewer |
Feedback from the Authors |
|
1. |
The paper was generally well written and can be accepted in the present form
|
Thank you for the comments and recommendations. |
Reviewer 3 Report
This reviewer thanks the authors for the submission and the Editor for the invitation to review it. The reviewer feels that the paper is interesting, and it is within in the scope of the Journal.
Specific comments:
- In section 1, The main issue is related with references, the citation in line 46 can be arranged in a more friendly way, i.e. thy can be placed in the same breckets. In line 52, the temperature value does not have the degree symbol correctly place. In line 57, the citation of reference 12 does not has the same name. in line 72, the citation of the team is not the best, please, remove by our team and leave the the name of Jumahat and co-workers, once more merge the references at the end of this statement. In line 76, please remove the symbol @. The inclusion of Table 1 on this section is not a good idea. In fact, all the information contained in Table 1 can be placed within the introduction text and, of course, this is not a review paper. Note that adding the table 1 information within the text can reduce at least one page.
- In section 3, on lines 172-174, change the verb were to are. In lines 185 and 186 authors speak of better compatibility of the nanosilica with the non-treated kenaf fibre, but it can also be due to the change of the performance of the treated kenaf fibre. Hence, authors should not make this statement without more prove, for instance, it would be interesting to compare composites with and without treated kenaf without the presence of nanosilica. In line 233, the word elsewhere is not appropriated, please, replace this word by another one like as reported by other researchers…
Author Response
Reviewer#3
|
No |
Comments/ Recommendations/ Suggestions/ Queries from the Reviewer |
Feedback from the Authors |
|
1. |
General Comments,
1. English language and style are fine/minor spell check required 2. Does the introduction provide sufficient background and include all relevant references? Must be improved 3. Comments and Suggestions for Authors
The reviewer feels that the paper is interesting, and it is within in the scope of the Journal.
|
All sections in the manuscript has been improved according to the reviewer’s comments.
|
|
2. |
In section 1,
1. The main issue is related with references, the citation in line 46 can be arranged in a more friendly way, i.e. thy can be placed in the same brackets. 2. In line 52, the temperature value does not have the degree symbol correctly place. 3. In line 57, the citation of reference 12 does not has the same name. 4. in line 72, the citation of the team is not the best, please, remove by our team and leave the the name of Jumahat and co-workers, once more merge the references at the end of this statement. 5. In line 76, please remove the symbol @. The inclusion of Table 1 on this section is not a good idea. In fact, all the information contained in Table 1 can be placed within the introduction text and, of course, this is not a review paper. Note that adding the table 1 information within the text can reduce at least one page.
|
In section 1,
1. In line 46, the citation all of references used has been placed in same brackets. (refer to page 2, in new line 53)
2. In line 52, the degree symbol has been corrected. (refer to page 2, in new line 59) 3. In line 57, the citation of reference 12 has been checked and corrected. (refer to page 2, in new line 64) 4. In line 72, the words “our team” and the references have been merged. (refer to page 2, in new line 80-81)
5. In line 76, the symbol @. has been removed. The table 1 has been removed. (refer to page 2, in new line 87)
|
|
3. |
In section 3,
1. on lines 172-174, change the verb were to are.
2. In lines 185 and 186 authors speak of better compatibility of the nanosilica with the non-treated kenaf fibre, but it can also be due to the change of the performance of the treated kenaf fibre. Hence, authors should not make this statement without more prove, for instance, it would be interesting to compare composites with and without treated kenaf without the presence of nanosilica. 3. In line 233, the word elsewhere is not appropriated, please, replace this word by another one like as reported by other researcher.
|
In section 3,
1. on lines 172-174, the verb “were” has been changed to are. (refer to page 5, in new line 199) 2. As suggested by reviewer, The SEM images (Figure 5) have been added. This figure is used to support the statement in line 185-186. (refer to page 5-7, in new line 208-276)
3. The word elsewhere has been changed with other researcher. (refer to page 8, in new line 329) |